# Magnetoelectric coupling in multiferroics probed by optical second harmonic generation

Shuai Xu [1,2,6], Jiesu Wang [3,6], Pan Chen [1,6], Kuijuan Jin [1,2,4] ✉, Cheng Ma [1,2], Shiyao Wu[3], Erjia Guo [1,2], Chen Ge [1,2], Can Wang [1,2,4], Xiulai Xu [1,5], Hongbao Yao [1,2], Jingyi Wang[1], Donggang Xie[1], Xinyan Wang[1,2], Kai Chang [3], Xuedong Bai [1,2,4] & Guozhen Yang[1,2]

Magnetoelectric coupling, as a fundamental physical nature and with the potential to add functionality to devices while also reducing energy consumption, has been challenging to be probed in freestanding membranes or two-dimensional materials due to their instability and fragility. In this paper, we report a magnetoelectric coupling probed by optical second harmonic generation with external magnetic field, and show the manipulation of the ferroelectric and antiferromagnetic orders by the magnetic and thermal fields in $BiFeO_3$ films epitaxially grown on the substrates and in the freestanding ones. Here we define an optical magnetoelectric-coupling constant, denoting the ability of controlling light-induced nonlinear polarization by the magnetic field, and found the magnetoelectric-coupling was suppressed by strain releasing but remain robust against thermal fluctuation for freestanding $BiFeO_3$.

Magnetoelectric (ME) coupling, generally existing in magnetoelectric[1,2] materials combining ferroelectric and magnetic behaviors, where the electric polarization can be manipulated by magnetic fields and the magnetization by electric fields, has potentially broad applications in spintronics, sensing, and energy harvesting technologies[2,3] and offers routes to design entirely new device architectures[2]. However, the probe and the control of these simultaneous ferroic orders become extremely challenging[2,4,5] for two-dimensional (2D) materials[6,7] or freestanding perovskites oxide films due to their instability and fragility extremely limiting the probe by traditional methods. Therefore, it is a desperately demand for the development of the technical and analyzing method, to characterize and to reveal the intrinsic mechanism of ME in low-dimensional materials, given their remarkable electronic properties[8], potential electronic applications[3,5,9], and many other functionalities to be explored.

Nonlinear optical susceptibilities of magnetic origin possess quite different transformation properties under space and time symmetry operations with nonlinear susceptibilities of electric origin[10]. Experimental data of optical second harmonic generation (SHG) allow us to clearly distinguish between time-invariant and time-noninvariant nonlinear susceptibilities[6,11,12]. Recently, rotational anisotropy SHG (RA-SHG) technology has been applied to study the antiferromagnetic order[6,7,10] and even ME coupling where the variation of the SHG asymmetry was obtained with an *electric* manipulation[12], however it remains insufficient to explore ME coupling[7]. As it will be presented in this paper, the ME coupling can be probed and manipulated quantitatively by the combination of wide temperature-range SHG (WT-SHG) and that with external *magnetic* field.

## Results

### Preparation of freestanding $BiFeO_3$ films

Bismuth ferrite is of particular interest because it is a typical room temperature multiferroic material with ferroelectricity ($T_c \approx 1100$ K), ferroelasticity, (anti)ferromagnetism ($T_N \approx 640$ K), and magnetoelectric

[1]Beijing National Laboratory for Condensed Matter Physics, Institute of Physics, Chinese Academy of Sciences, 100190 Beijing, China. [2]University of Chinese Academy of Sciences, 100049 Beijing, China. [3]Beijing Academy of Quantum Information Sciences, 100193 Beijing, China. [4]Songshan Lake Materials Laboratory, 523808 Dongguan, Guangdong, China. [5]State Key Laboratory for Mesoscopic Physics and Frontiers Science Center for Nano-optoelectronics, School of Physics, Peking University, 100871 Beijing, China. [6]These authors contributed equally: Shuai Xu, Jiesu Wang, Pan Chen. ✉e-mail: kjjin@iphy.ac.cn

coupling effects[13–17]. The large-scale freestanding BiFeO₃ (BFO) films presented here were obtained by releasing the BFO films from the substrates, i.e., through dissolving the sacrificial layer in de-ionized water[18], with BFO films epitaxially grown on about 17-nm-thick water-soluble sacrificial $Sr_3Al_2O_6$ (SAO) which was fabricated on (001)-oriented $SrTiO_3$ (STO) substrates, and then transferring to any other solid substrate or flexible one, schematically shown in Fig. 1a. The high quality of BFO/STO and BFO/SAO/STO heterostructures, as well as that of freestanding BFO films was confirmed by X-ray diffraction (XRD), reciprocal space mapping (RSM) (Supplementary Fig. 1), the high-resolution spherical aberration-corrected transmission electron microscopy (TEM) characterization, and by atomically resolved energy-dispersive X-ray spectroscopy (EDS) mapping of the Bi, Fe, Sr, and Al elements (Supplementary Fig. 2). The 5 mm × 5 mm freestanding BFO films on the polydimethylsiloxane (PDMS) exhibit high flexibility. Only shift but no splitting of (103) and (013) RSM peaks indicate the coherently growth, and smooth surface at atomic level were characterized by atomic force microscopy for the freestanding BFO films (Supplementary Fig. 1). From the HAADF-STEM images and fast Fourier transform (FFT) patterns on the upper surface and at the interface of BFO grown on STO (Supplementary Fig. 3) respectively, we find that the in-plane lattice constant is basically the same (3.89 Å) on the upper surface with that near the interface, while the out-of-plane lattice constant is a little smaller

(4.02 Å) on the upper surface than that (4.09 Å) close to the interface between BFO and the substrate of STO. Although there is some relaxation of the compressive stress from the substrates for the lattices on the upper surface of BFO films with the thickness of 47 nm epitaxially grown on the STO, the stress was well kept within the films concluded from the larger out of plane lattice constant than that (3.979 Å) of freestanding BFO (Supplementary Fig. 1a). After releasing the stress of SAO/STO, the out-of-plane lattice constant of the freestanding BFO films became smaller (from 4.061 to 3.979 Å).

## Ferroelectric properties of BiFeO₃ films

The ferroelectric properties of BFO/STO, BFO/SAO/STO, and freestanding BFO were obtained by the piezoelectric force microscopy (PFM) and the high-resolution spherical aberration-corrected TEM. In order to compare the polarization strength of different forms of BFO films, all BFO films were grown with the same thickness of about 47 nm. From the measurements of XRD (Supplementary Fig. 1a), RSM (Supplementary Fig. 1g–l), FFT patterns (Supplementary Fig. 3), and RA-SHG, it can be concluded that: with the thickness of 47 nm, the stress on the BFO films from the substrates was well kept, so that the epitaxial BFO films on STO and those on SAO/STO were in a tetragonal-like (T-like) phase and most likely with a $P4mm$ space group, consistent with our previous study[19], while it was also feasible to obtain the large-

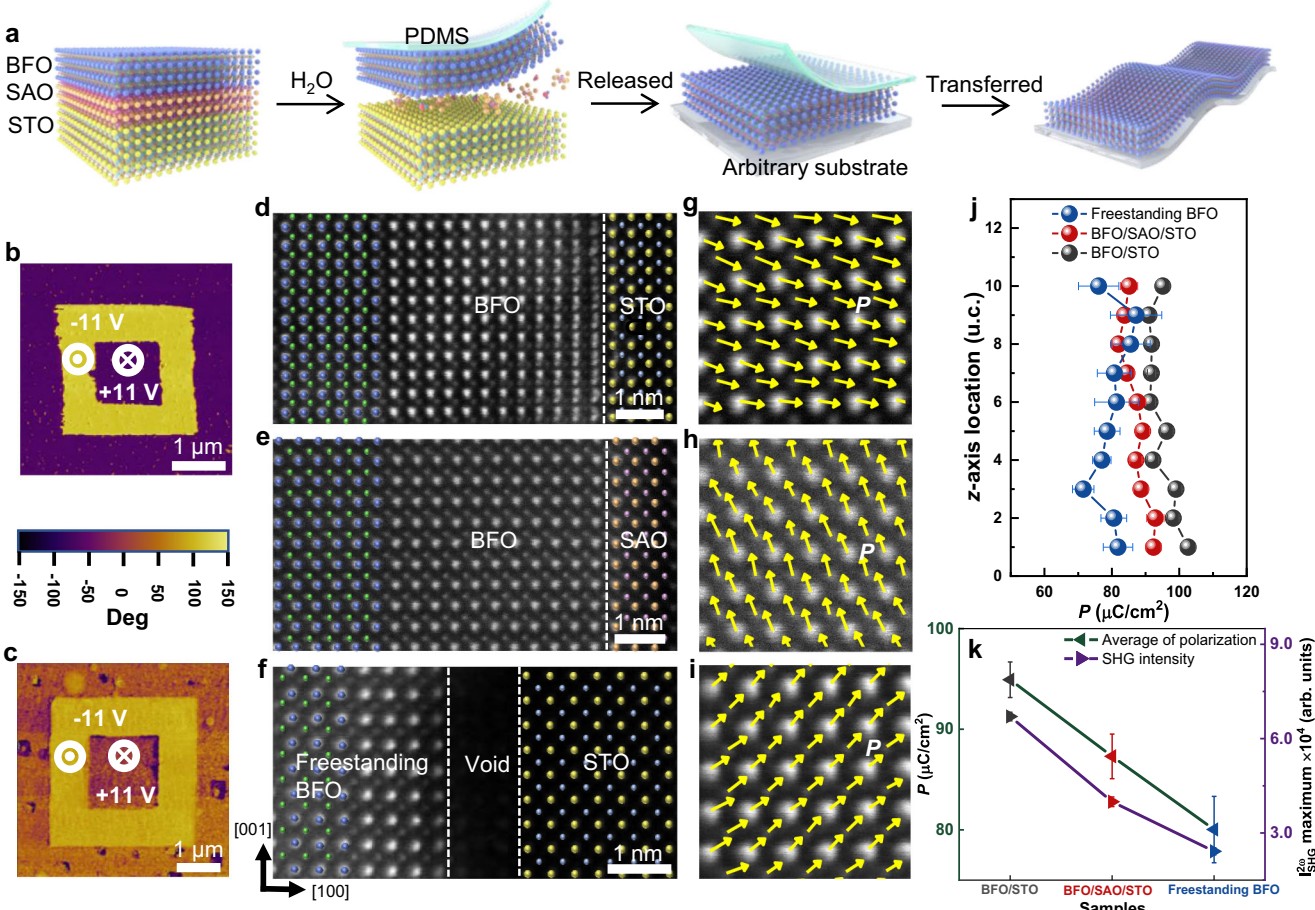

**Fig. 1 | Preparation and ferroelectric properties characterization of BFO films. a** Schematics of the whole fabrication process: from BFO/SAO/STO heterostructures to freestanding BFO films. **b, c** Out-of-plane PFM phase images of the epitaxial BFO films and the freestanding BFO films, respectively. **d–f** The cross-sectional STEM-HAADF images of BFO/STO, BFO/SAO/STO, and freestanding BFO transferred on STO, respectively. **g–i** The visualization of polarization in the corresponding films in **d–f**. The yellow arrows are plotted according to the

displacement of Fe relative to the Bi sublattices, the position of which are determined by the two-dimensional Gaussian algorithm. **j** The average polarization magnitude calculated by the empirical equation. The error bar represents the standard deviation of measured unit-cells. **k** Comparison of the average polarization and the maximum value of SHG intensity for BFO/STO, BFO/SAO/STO, and freestanding BFO films at room temperature. The error bar represents the standard deviation of measured samples.

scale freestanding BFO film which stayed in a rhombohedral-like (R-like) phase and with an $R3c$ space group. The nearly 180° phase contrast of PFM images for the BFO films epitaxial grown on STO with SrRuO$_3$ as conducting buffer layer and the much smaller one for the freestanding BFO (Fig. 1b, c), showing the reversibility of the ferroelectric polarization, indicate the good ferroelectric property in BFO films epitaxial grown on substrates and a weaker one in the freestanding BFO films. The mapping of the polarization configuration for three kinds of samples (Fig. 1g–i) was demonstrated by calculating the displacements of Fe and Bi sub-lattices using a two-dimensional Gaussian fitting algorithm[20,21] based on the scanning TEM high-angle annular dark-field (STEM-HAADF) images (Fig. 1d–f), which was acquired by adding up a three-image stack from the same region to minimize the sample drifts and scan noise effects.

To quantitatively explain the polarization evolution in these three samples, analysis at unit cell scale is needed. An empirical linear relationship[22,23] between the polarization strength with respect to the offset between the Fe cation and the four surrounding Bi cations was adopted to obtain the polarization semi-quantitatively for BFO/STO, BFO/SAO/STO, and freestanding BFO films. The profile of polarization (Fig. 1j), also denoted by the yellow vectors in Fig. 1g–i, suggests that the maximum amplitude observed in BFO/STO is about 94.92 μC/cm$^2$, which is consistent with the results of 60–100 μC/cm$^2$ reported in the literature[24–26], and it slightly decreased to about 87.30 μC/cm$^2$ with the SAO buffer layer, while the minimum one appeared in the freestanding BFO films is about 80.03 μC/cm$^2$. The reduction of polarization amplitude in BFO/STO, BFO/SAO/STO, and freestanding BFO, consistent with the results reported in the literature[27], is due to the sequential relaxation of the strain. The polarization magnitude calculated from TEM method is in good agreement with that from SHG (Fig. 1k), confirming the validity of our results.

## Antiferromagnetic phase transitions probed by SHG

The ferroelectric properties of BFO/STO, BFO/SAO/STO, and freestanding BFO were obtained by the measurement of optical SHG which was ensured by its linear dependence with the square of the incident optical power (Supplementary Fig. 4). It has been found that the ferroelectric polarization in the BFO films directly coupled with the non-collinear G-type antiferromagnetic as well as the weak ferromagnetic moment driven by the Dzyaloshinskii–Moriya (DM) interaction, which arises from spin–orbital coupling in antisymmetric systems[17,28–31]. The SHG intensity $I(2\omega)$ is related to the light-induced nonlinear polarization $\boldsymbol{P}(2\omega)$ in the following way: $I \propto |\boldsymbol{P}|^2$, where $\boldsymbol{P}(2\omega) = \varepsilon_0(\chi^{(i)} + \chi^{(c)}) : \boldsymbol{E}(\omega) \otimes \boldsymbol{E}(\omega)$, with $\boldsymbol{E}(\omega)$ denoting the incident light electric field, $\chi^{(i)}$ and $\chi^{(c)}$ as the time-invariant and time-noninvariant SHG tensors (see part 12 in Supplementary Information for details), associated with the crystallographic (ferroelectric) and G-type antiferromagnetic order, respectively[11,12]. To characterize the evolution of antiferromagnetic order, ferroelectric order, and the ME coupling with the manipulation of stress, temperature, and applied magnetic field, the measurements of wide temperature-range RA-SHG and that with applied magnetic field were carried out (Fig. 2).

The SHG setup is with a reflection geometry for all SHG measurements as sketched in Fig. 2a. Firstly, an abrupt increase of the SHG intensity at 618 K was observed in BFO/STO with the WT-SHG measurements from 750 to 200 K (Fig. 2b). By extracting the ferroelectric and the G-type antiferromagnetic orders from the RA-SHG results at different temperatures (Fig. 2c and Supplementary Fig. 5), corresponding to the time-invariant $\chi^{(i)}$ and time-noninvariant tensors $\chi^{(c)}$, respectively, the contribution of SHG signal from these two orders, as well as the reduction of them with increasing temperature were obtained and shown in Fig. 2d–f. The temperature-dependent SHG signal (extracted from Fig. 2c–f and Supplementary Fig. 5) contributed by ferroelectric order gradually decreases with the increase of temperature (Supplementary Fig. 6), indicating that the ferroelectric

properties are weakening with the increase of temperature, which is consistent with the ferroelectric order parameter variation studied by XRD[13]. The first-order phase transition at Néel temperature ($T_N \approx 618$ K) was clearly observed (Fig. 2g), above which the antiferromagnetic feature completely vanished. Here, a well-known phenomenological function[6,7] $I(2\omega) \propto [a + b(T_N - T)^\beta]^2$, was used to fit our data for the antiferromagnetic-paramagnetic phase transition. We believe that the asymmetry of the wide temperature-range RA-SHG patterns at the temperature lower than 618 K for BFO/STO (Supplementary Fig. 5), together with that observed at room temperature for the freestanding BFO films (Supplementary Fig. 7) is a distinguishing feature for the existence of time-noninvariant contribution, where the possibility of the coexisting of two crystal structures[32] in BFO has been eliminated.

The RA-SHG results on the freestanding BFO films with the manipulation of in-plane magnetic field from −6 to 6 T at 10 K (Fig. 2h) and those at the room temperature (Supplementary Fig. 7) clearly demonstrate the suppression of SHG signals generated by the ferroelectric order and antiferromagnetic order by the external magnetic field. The magnetoelectric coupling in BFO was induced by its intrinsic and significant spin-orbital coupling. The antiferromagnetic order would introduce additional electric polarization via spin-orbital coupling directly, which is well-known as magneto-striction phenomenon[33,34]. Furthermore, the coupling between the ferroelectric order and the non-collinear G-type antiferromagnetic order, which is induced by the DM interaction, offers a more complex way to manipulate the electric property and magnetic property by each other[35,36]. That is, the reduction of the ferroelectric order contributed to the SHG (Fig. 2i–k) can be attributed to the orientation varying of the electric polarization[37]. Nevertheless, we think whether and how (if yes) the magnetic field affects the strength of the polarization for BFO is still an open question and need to be further studied by all means. The coupled SHG signal from the non-collinear G-type antiferromagnetic order and ferroelectric order clearly shows the robustness of the ME coupling in freestanding BFO even with the applied magnetic field of ±6 T not only at 10 K, but also at the room temperature. This high temperature ME coupling characteristics of freestanding BFO films may increase their potential applications for the multifunctional 2D device in the future.

To further understand the mechanism for the spatial-inversion symmetry breaking caused by the antiferromagnetic order, the electronic dipole moment and ionic dipole moment were calculated via modern polarization theory[38] as implied in the Vienna ab initio Simulation Package[39,40]. As shown in Supplementary Table 1 and Supplementary Fig. 8, for all three components, the dipole moments of BFO with G-type antiferromagnetic order were enhanced compared to those with FM order. The enhanced electric polarization can be attributed to the G-type antiferromagnetic order induced spatial-inversion symmetry breaking by spin-orbital coupling effect.

## Magnetoelectric coupling revealed by SHG

To manipulate and control ME coupling more accurately and reveal more insight of its mechanism, systematic measurement of SHG and that of magnetic moment with applied in-plane magnetic field have been accomplished for BFO/STO, BFO/SAO/STO, and freestanding BFO films. The saturation magnetization moment of BFO/STO is about 8 emu/cm$^3$, identical with that in other reports[41]. The saturation magnetization moment of freestanding BFO films was significantly increased about 7 times (3 times) comparing with that of BFO/STO (BFO/SAO/STO) (Fig. 3a), while the ferroelectric and antiferromagnetic characteristics were both weakly reduced, which is consistent with the conclusion that ferromagnetism and ferroelectricity are generally mutually exclusive[42]. We think that the antiparallel spins in the antiferromagnetic order are more close to pseudo-collinear antiferromagnetic order in BFO with the cycloidal propagation direction $\boldsymbol{k}$

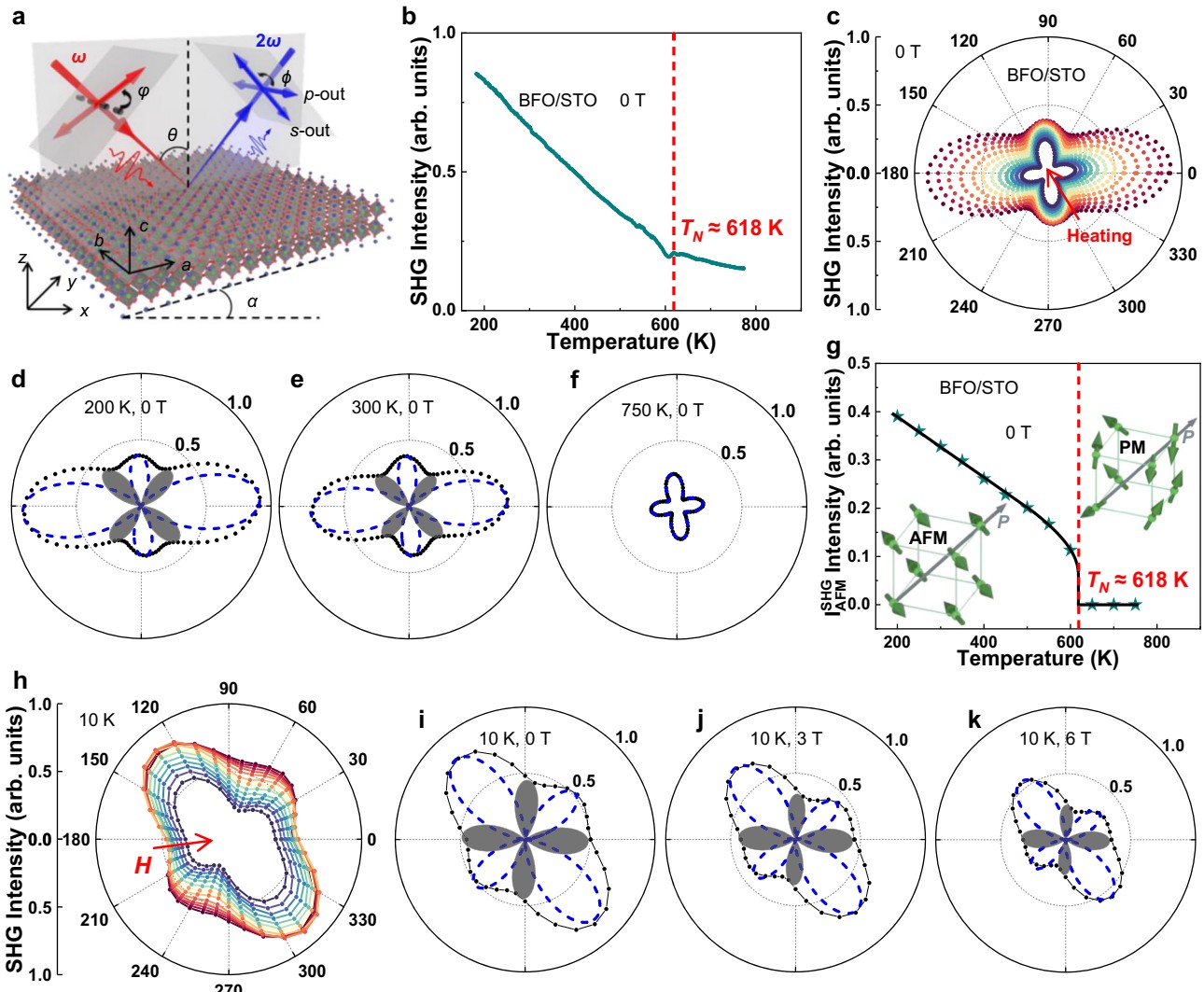

**Fig. 2 | Wide temperature-range (RA-) SHG and that with applied magnetic field (*H*) measurements. a** Schematic diagram of the polarization configuration in the reflective geometric light path. **b** The SHG intensity of BFO films as a function of temperature in the *p-p* polarization configuration. **c** RA-SHG measurements of BFO/ STO films at different temperatures, the scattered points are the experimental data at different temperatures. **d–f** Theoretical fitting results of RA-SHG measurements of BFO/STO films at 200, 300, and 750 K, respectively. The scattered points are experimental data, the blue dashes represent the contribution of the ferroelectric order to the SHG signal, and the gray shaded areas represent the contribution of the

antiferromagnetic order to the SHG signal. **g** Temperature dependent SHG signals contributed by antiferromagnetic order extracted from RA-SHG measurements. Insets: Schematic diagrams of antiferromagnetic order and paramagnetic order. **h** RA-SHG of freestanding BFO films under different magnetic fields. **i–k** Theoretical fitting results of RA-SHG measurements of freestanding BFO films at 0, 3, and 6 T, respectively. The dotted lines are experimental data, the blue dashes represent the contribution of the ferroelectric order to the SHG signal, and the gray shaded areas represent the contribution of the antiferromagnetic order to the SHG signal.

along $[\bar{2}11]$, $[1\bar{2}1]$, and $[11\bar{2}]$, corresponding to BFO/STO with larger strain in $ab$ plane and larger $c$, than those with the **k** along $[1\bar{1}0]$, $[10\bar{1}]$, and $[01\bar{1}]$, corresponding to the freestanding BFO without strain and with smaller $c$ (schematically shown in Fig. 3b, c)[43–45]. Therefore, the strongest (or weakest) antiferromagnetic order in BFO/STO (or freestanding BFO) (Fig. 3f–h), as well as the weakest (or strongest) magnetic feature caused by the G-type antiferromagnetic cycloidal order, can be well explained and understood[46].

The significant enlargement of the saturation magnetization moment could also be attributed to the weakening of the electric polarization and the variation of the related DM interaction (schematically shown in Supplementary Fig. 9a) in the following two aspects: The first one is the residual magnetic moment enlargement resulted from the variation of the period of the antiferromagnetic cycloidal order (Supplementary Fig. 9b, c)[47,48], and the second one is that the effective magnetic moment of the antiferromagnetic cycloidal order

could be enhanced due to the transition of propagation direction[45], so as the residual magnetic moment (Supplementary Fig. 9c, d). The smaller magnetic moment we observed in the BFO/STO films and the much smaller enhancement (only 2.7 times increasing) in the freestanding one with the thickness of 53 nm (Supplementary Fig. 10) confirmed this mechanism to a certain extent and was consistent with the observation reported by Huang et al.[48]. Nevertheless, further systematic study on this issue is highly expected in the future.

To see the evolution of the ferroelectric order and the antiferromagnetic order with the strain releasing, the RA-SHG results of BFO/STO, BFO/SAO/STO, and freestanding BFO films at room temperature without magnetic field are shown in Fig. 3f–g. Combining with the results shown in Fig. 2, we can conclude that the strongest (or weakest) ferroelectric order in BFO/STO (or freestanding BFO) can be obtained while the antiferromagnetic order is the strongest (or the weakest).

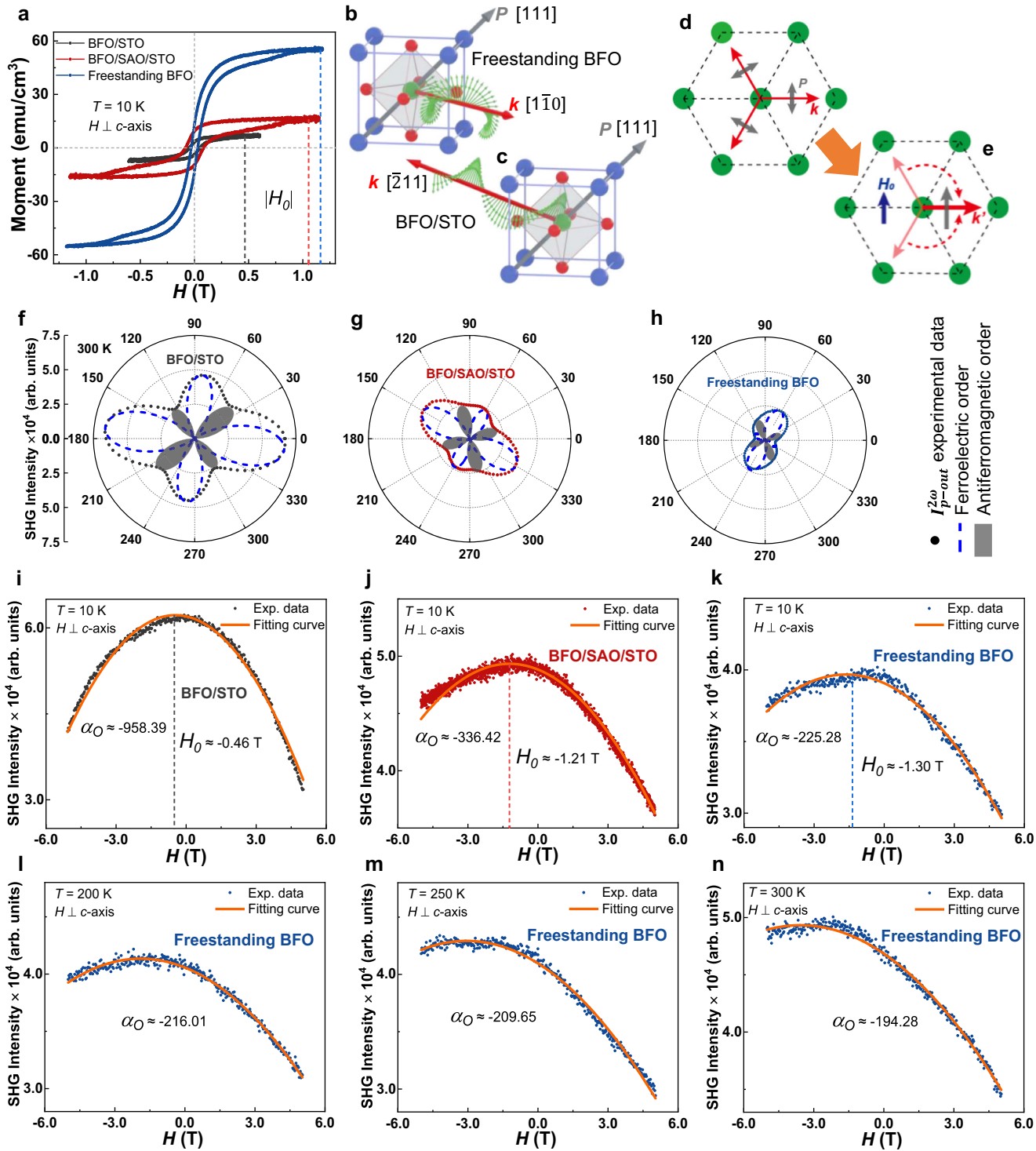

**Fig. 3 | SHG with applied magnetic field and M-H measurements. a** M-H diagrams of BFO/STO, BFO/SAO/STO, and freestanding BFO films transferred onto STO (001) substrates. **b**, **c** Schematic diagrams of the non-collinear G-type cycloid antiferromagnetic order in the freestanding BFO films with low strain and that in BFO/STO films with high strain, respectively. **d** Schematic diagram of the relationship between polarization **P** and cycloidal vectors **k** without the magnetic field. **e** Schematic diagram of the relationship between polarization **P** and cycloidal vectors **k'** with the magnetic field. **f–h** Room temperature RA-SHG measurements of BFO/STO, BFO/SAO/STO, and freestanding BFO films without magnetic field, respectively. **i–k** Magnetic field-dependent SHG of BFO/STO, BFO/SAO/STO, and freestanding BFO films in the *p-p* polarization configuration, respectively. **l–n** Magnetic field dependent SHG of freestanding BFO films at 200, 250, and 300 K, respectively.

The SHG results with applied magnetic field (from −5 to 5 T) of BFO/STO, BFO/SAO/STO, and freestanding BFO at the low temperature of 10 K, as well as those at the temperature between 200 and 300 K for freestanding BFO, clearly show an upset down parabolic behavior (Fig. 3i–n). The magnetic field-dependent SHG signal

(extracted from Fig. 2h–k) contributed by antiferromagnetic order and ferroelectric order also exhibits an upset down parabolic behavior (Supplementary Fig. 11). A quadratic function $I = \alpha_O(\mathbf{H} + H_0)^2 + I_0$ was used to fit the experimental data, where $I_0$ is the strongest intensity with the applied magnetic field of $H_0$, and $\alpha_O$ denotes the magnetic

manipulating capacity for the ferroelectric order. As $|\boldsymbol{P}|^2 \propto I$, we can obtain the following expression:

$$|\boldsymbol{P}|^2 \propto \alpha_O(\boldsymbol{H} + H_0)^2 + I_0, \tag{1}$$

here we define $\alpha_O$ as an optical magnetoelectric-coupling constant, related to the strength of ME coupling and the controlling ability of light-induced nonlinear polarization by the applied magnetic field. This constant $\alpha_O$ is obtained as −958, −236, and −225, for BFO/STO, BFO/SAO/STO, and freestanding BFO, respectively, and is found to be the largest (absolute value) in the system with the largest ferroelectric order and the strongest antiferromagnetic order simultaneously, indicating the weakening of the magnetoelectric coupling order resulted from the stress releasing. Unsurprisingly, we found $H_0$ in the expression (1) corresponds to the magnetic field with which the saturation magnetization moment was obtained. We think the three propagating directions $\boldsymbol{k}$ rotated to one single direction $\boldsymbol{k}'$ by the magnetic field $H_0$ when the magnetization was saturated (Fig. 3d–e), meanwhile the direction of the electric polarization $\boldsymbol{P}$ also tends to align in one single direction perpendicular to $\boldsymbol{k}'$ due to the existence of the DM interaction[35]. This is the reason why the strongest SHG signal was obtained at the same magnetic field $H_0$ as the saturation filed shown in Fig. 3a. Although the slightly decreasing of $|\alpha_O|$ with the increasing of the temperature (-216 at 200 K, 210 at 250 K, and 194 at 300 K) was observed in the freestanding BFO, the ME coupling robustly remained with the same order of $\alpha_O$ at the temperature of 300 K (Fig. 3l–n), indicating a great manipulating capacity of either magnetic or electric property by the other filed and its potential application for the multifunctional device in the future.

## Discussion
In summary, taking model multiferroic BFO as an example, we report a ME coupling probed by WT-SHG with external magnetic fields. We demonstrate the systematic evolution of ferroelectric and anti-ferromagnetic orders as a function of applied magnetic field and temperature for differently strained BFO films, especially for freestanding BFO.

We define an optical magnetoelectric-coupling constant, denoting the ability of controlling light-induced nonlinear polarization by the magnetic field, and find that the magnetoelectric coupling was suppressed by strain releasing, as well as the antiferromagnetic and ferroelectric orders, but remain robust against thermal fluctuation for freestanding BFO films. We also find that this optical ME coupling constant remains in the same order in freestanding BFO with those in the films grown on the substrates, indicating the robustness of ME coupling against strain releasing. The first-order phase transition with the Néel temperature of 618 K in BFO films and an -7 times enlargement of saturate magnetization moment in freestanding BFO were also observed, and the later was attributed to the variation of DM interaction. The robust ME coupling in the freestanding BFO against thermal fluctuation suggests the potential application for multifunctional devices in the future. We believe that the demonstrated advanced SHG technology with tunable external fields pave a unique way to investigate the magnetoelectric coupling and antiferromagnetic order for other freestanding multiferroic films or 2D materials.

## Methods
### Synthesis of samples
Both SAO and BFO films were deposited on (001)-oriented STO substrates by pulsed laser deposition technology using a XeCl excimer laser with a wavelength of 308 nm. The water-sacrificial SAO layer was grown at 780 °C under an oxygen pressure of 2.0 Pa with a substrate-target distance of 7.5 cm, the laser energy density of -2.2 J/cm², and the repetition rate of 2 Hz. The BFO layer was subsequently deposited at 700 °C under an oxygen pressure of 20.0 Pa with a substrate-target

distance of 7.5 cm, the laser energy density was -1.6 J/cm², and the repetition rate of 2 Hz. After the growth, the heterostructures were annealed in situ at the grown conditions of BFO layer for 10 min to maintain surface stoichiometry and then was cooled down to room temperature with a rate of 25 °C/min.

### Release and transfer BFO films
A 10 mm × 20 mm × 0.5 mm transparent and flexible PDMS was tightly covered on the surface of the BFO/SAO/STO (001) epitaxial films. They were then immersed in de-ionized water at room temperature for about 30 min until the SAO sacrificial layer was completely dissolved and the BFO films were separated from the STO (001) substrate. After that, the PDMS together with the freestanding BFO films were dried with N₂ gas for several minutes. Then, it was transferred onto any desired substrate (such as a silicon wafer or a TEM grid), with the entire stacking annealed at 90 °C for 30 min to promote adhesion. After cooling to 70 °C and slowly peeling off the PDMS with tweezers, the transferred films on an arbitrary substrate were obtained.

### Structural and basic physical properties characterizations
XRD, X-ray reflectivity (XRR), and RSM were carried out using a Panalytical X'Pert3 MRD diffractometer with Cu-$K\alpha_1$ (1.54056 Å) radiation equipped with a 3D pixel detector. The macroscopic magnetization measurements of all samples were performed using a vibrating sample magnetometer (VSM) with a Physical Property Measurement System (PPMS) operating at a vibration frequency of 40 Hz and a vibration amplitude of 2 mm with an applied in-plane magnetic field. M-H hysteresis loops were recorded at 10 K with the magnetic fields ranging from −1.5 to +1.5 T. The detailed process is as follows. First, the M-H hysteresis loops of BFO/STO and BFO/SAO/STO films were measured. Then, the BFO/SAO/STO films were immersed in deionized water to obtain the freestanding BFO films. After that, the freestanding BFO films were transferred to (001)-oriented STO substrates and put into PPMS again for M-H measurements.

### TEM sample preparation and data acquisition
All the TEM specimen, including the STO/BFO, STO/SAO/BFO and freestanding BFO on STO, were made by focused ion beam process using FEI Scios 2 dual-beam system. The STEM-HAADF images were taken on a JEOL ARM 300 F microscope at 300 kV with a convergence angle of 18 mrad and a collection angle of 54–220 mrad. The displacements of the Fe sublattice relative to the Bi sublattice were calculate by the Matlab program, which determined the positions of Bi and Fe atoms by a two-dimensional Gaussian fitting algorithm. The polarization of BFO films were obtained from an empirical equation: $\boldsymbol{P}_s = C \cdot \Delta\boldsymbol{d}$, where $\boldsymbol{P}_s$ is the spontaneous polarization, $C$ is a constant, and $\Delta\boldsymbol{d}$ is the relative displacement. As $C$ can be obtained from ref. 23, we can calculate polarization easily.

### Optical SHG measurements
The room temperature RA-SHG, WT-SHG, and SHG with applied magnetic field measurements of BFO films were all performed in a typical reflection geometry, as shown in the schematic diagram of Fig. 2a. The incident laser beam was generated by the Maitai SP Ti:Sapphire oscillator produced by Spectra Physics, which produces a femtosecond pulsed laser with the incident light power kept at 50 mW and a center wavelength of 800 nm (pulse width 120 fs, frequency 82 MHz). Both the incident angle and the reflection angle are fixed at 45°, and the polarization direction $\varphi$ of the incident light field is adjusted by the rotation of the $\lambda/2$ waveplate driven by a rotating motor. The second harmonic signal is collected by a photomultiplier tube and transmitted to a photon counter. The polarization configuration of the reflection light was fixed as $p$ or $s$ polarization, and the rotated anisotropy patterns under different reflection polarization configurations were obtained by rotating the incident light polarization angle $\varphi$.

The temperature-variable stage used in the WT-SHG measurements process is a Heating And Cooling Stage, i.e., model HFS600E-PB4 produced by LINKAM, UK, with a long temperature-range of 77–873 K. The change in SHG intensity was measured from 200 to 750 K with the polarization configurations of the incident and reflection light fixed both $p$ and $p$ polarization. The wide temperature-range RA-SHG is measured every 50 K from 200 to 750 K. The SHG with applied magnetic field measurements were performed based on cryogenic optical research instrument (OptiCool, Quantum Design). The surrounding SHG system is self-built. With the polarization configurations of the incident and reflection light fixed as $p$ and $p$ polarization, the temperature set at 10 K, and the changing magnetic field (from −5 to +5 T) applied along the in-plane direction of the samples, the magnetic-field-dependent SHG intensity of the BFO films can be obtained. Besides, the RA-SHG was measured with holding the magnetic field at every integer tesla from −6 to +6 T.

### First-principles calculations

Our first-principles calculations were based on density-functional theory as implemented in the Vienna ab initio simulation package[40,49], using the projector augmented-wave method[50]. The exchange–correlation potential was adopted by the generalized gradient approximation (GGA) of the Revised PBE for solids[51]. The GGA + U method[52] was adopted to improve the description of on-site Coulomb interactions of the Fe-3$d$ orbitals with the effective $U_{eff}$ set as 4.0 eV. The plane-wave cutoff energy was set to 520 eV. A $2 \times 2 \times 2$ supercell was adopted to describe the antiferromagnetism and the polarization.

### Data availability

The data that support the findings of this study are available on the proper request from the first author (S.X.) and the corresponding author (K.J.).

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

## Acknowledgements

This work is supported by the National Key Basic Research Program of China (Grant Nos. 2019YFA0308500, 2020YFA0309100, and 2021YFA1400701), the National Natural Science Foundation of China (Grant Nos. 11721404, 11934019, 11974390, 12104054, 12074038, 92165104, 12222414, and 12074416), the Youth Innovation Promotion Association of the Chinese Academy of Sciences (Grant Nos. 2018008 and Y2022003), the Beijing Nova Program of Science and Technology (Grant No. Z191100001119112), Beijing Natural Science Foundation (Grant Nos. 2202060 and 1222035), and the Strategic Priority Research Pro-gram (B) of the Chinese Academy of Sciences (Grant No. XDB33030200).

## Author contributions

These samples were grown and processed by S.X. under the guidance of K.J.; wide temperature-range (RA-) SHG measurements were conducted by S.X. under the guidance of K.J.; optical (RA-) SHG with applied magnetic field measurements were conducted by Jiesu Wang, S.W, and K.C.; TEM lamellas fabrications and TEM experiments were performed by P.C. directed by X.B.; theoretical calculations were performed by C.M. under the guidance of K.J.; part of the schematics was performed by D.X; PFM experiments were performed by X.W. and S.X.; S.X. worked on the structural and magnetic measurements. Jiesu Wang, C.M., E.G., C.G., C.W., X.X., G.Y., H.Y., and Jingyi Wang participated the discussions and provided important suggestions during the manuscript revision. K.J. initiated the research and supervised the work. K.J., S.X., and C.M. wrote the manuscript with inputs from all authors.

## Competing interests

The authors declare no competing interests.

## Additional information

**Peer review information** *Nature Communications* thanks Manrong Li, Gaokuo Zhong, and the other anonymous reviewer(s) for their con-tribution to the peer review of this work. Peer reviewer reports are available.

