## [Peer Review File · Nature Communications]

Magnetoelectric Coupling in Multiferroics Probed by Optical Second Harmonic GenerationREVIEWER COMMENTS

Reviewer #1 (Remarks to the Author):

The paper reports important phenomena related to magneto-electric coupling in a ferroelectric material system by probing magnetic field dependence of the second harmonic generation,

The study presents convincing results, but the authors need to discuss the possibility of magneto-striction in addition to magneto-electric effects influencing the SHG. After adding this discussion the paper would be stronger.

Reviewer #2 (Remarks to the Author):

The authors report magnetoelectric coupling phenomena probed by wide temperature-range second harmonic generation (WT-SHG), and show the manipulation of the ferroelectric and antiferromagnetic orders in epitaxial and freestanding BiFeO₃ films. Developing RA-SHG technique is a unique way to investigate magnetoelectric coupling for other freestanding multiferroic films or 2D materials. The manuscript has the potential to be published in NC, but there are still many issues in the present form. The authors are suggested to respond following comments/questions:

1. The authors need to explain why they chose BFO film with a thickness of 47nm? BFO has special phase at such thickness? BFO has a special stress state at such thickness? BFO has greater polarization or larger optical second harmonic effects at such thickness?
2. The authors calculated the polarization magnitude of BFO/STO, BFO/SAO/STO, and freestanding BFO from HAADF images in Fig. 1, and they made a simple comparison among these three samples. The authors should quantitatively explain the polarization magnitude for these three samples, and compare the polarization intensity with reported literature.
3. In Fig. 2b and Fig. 2g, the authors present the SHG intensity of BFO films as a function of temperature and temperature dependent SHG signals contributed by antiferromagnetic order respectively. The temperature dependent SHG signals contributed by ferroelectric order should be provide and analysis.
4. The authors claim that “the strongest (or weakest) ferroelectric order in BFO/STO (or freestanding BFO) can be obtained while the antiferromagnetic order is the strongest (or the weakest), based on the results shown in Fig. 3f-g and Fig. 2”. For comparison, the authors are suggested to provide temperature-dependent SHG signals contributed by ferroelectric order and magnetic field-dependent SHG signals contributed by antiferromagnetic order for BFO/STO, BFO/SAO/STO, and freestanding BFO films.
5. How does the ferroelectric as a function of applied magnetic field for differently strained BFO films?
6. As the magnetoelectric-coupling was suppressed by strain releasing, as well as the antiferromagnetic and ferroelectric orders, how does the SHG patterns change under strain manipulate for freestanding BFO films?

Reviewer #3 (Remarks to the Author):

The authors intensively studied the magnetoelectric feature of thin-film BiFeO₃ in both strained and freestanding film forms, SHG is applied to intrinsically probe their tuned multiferroics under multiple coupling of magnetism, polarization, and optical response. These findings are expected to unveil the magnetoelectric coupling mechanism of related materials.

1. The symmetry of film form BiFeO₃ is not addressed. Is that the rhombohedral R3c or tetragonal P4mm? Although all can be related to the pseudo-cubic structure, it is necessary to clarify this since different polymorphs of BFO have been reported in thickness/substrate dependent cases.
2. The cell evolution indicates that the lattice mismatch between the substrate and BFO introduce tensile stress to BFO since the dimension along the c-axis become smaller in freestanding state, says from 4.061 to 3.979 Å? Is this for the unit cell parameter c of the P4mm symmetry? Or the d value for certain (hkl)?
3. As far as I understood, structurally, the freestanding and strained BFO should be considered as different phases due to their large dimensional difference, since the magnetostriction-polarization coupling is commonly observed in related system, such as in Mn₂MnWO₆ (Nat. Commun. , 2017, 8, 2037). So the multiple coupling effect cannot be simply counted when spin-lattice-phonon-dipole are got involved. So it could be a plus to measure any samples before and after removal of the intermediate Sr-Al-O layer for comparison.
4. The SHG was detected by reflection mode. In the strained BFO film, the out-plane stress reaches maximum in the BFO and substrate interface, being significantly faded with growing film thickness. My question is, for the ~50 nm thickness film, what is the stress effect on the upper surface compared with the freestanding one? So it is better to evaluate strain effect on surfaces of both strained and freestanding (top and bottom) ones by the lattice dimension. Presumably, the thicker the film, the smaller the lattice dimension of the outer-plane, the limit should be equal to that of the freestanding case.
5. Is the freestanding film totally relaxed? Any SHG difference measured on the top and bottom (the side epitaxially grown on substrate) surfaces?
6. The SHG-magnetism-polarization-lattice coupling is rather complicated. The authors are brave enough to touch this. It will be better to calculate the spontaneous polarization according the cell-parameter-based cif files by either first-principles calculations or point-charge model, so that one can tell the structurally dipole contribution.
7. The properties of these kind of films are very sensitive to defect, such as oxygen vacancy. Any characteristics on this – says how reproducible of the samples from different batches?

RESPONSE TO REVIEWER COMMENTS

Manuscript ID: NCOMMS-23-01445-T

Title: Magnetoelectric Coupling in Multiferroics Probed by Optical Second Harmonic Generation

We thank all the reviewers for positive recommendation and valuable comments regarding our research paper. Each of your insights have served to strengthen our manuscript. We have carefully revised the manuscript according to your constructive suggestions. Provided below is our detailed response to each comment raised.

Reviewer #1:

Comment. The paper reports important phenomena related to magneto-electric coupling in a ferroelectric material system by probing magnetic field dependence of the second harmonic generation. The study presents convincing results, but the authors need to discuss the possibility of magneto-striction in addition to magneto-electric effects influencing the SHG. After adding this discussion the paper would be stronger.

Response. *We greatly appreciate the positive comments from this referee. We thank the reviewer for the constructive suggestions that are important for the improvement of the manuscript. We have addressed the reviewer's suggestion in the following and in the revised manuscript.*

In the revised manuscript on page 3-4, we have added the relevant description as: "It has been found that the ferroelectric polarization in the BFO films directly coupled with the non-collinear G-type antiferromagnetic as well as the weak ferromagnetic moment driven by the Dzyaloshinskii-Moriya (DM) interaction, which arises from spin-orbital coupling in antisymmetric systems^{17, 28, 29, 30, 31}".

On page 5, we have rewritten the relevant description as: "The magnetoelectric coupling in BFO was induced by its intrinsic and significant spin-orbital coupling. The antiferromagnetic order would introduce additional electric polarization via spin-orbital coupling directly, which is well-known as magneto-striction phenomenon^{33,34}. Furthermore, the coupling between the ferroelectric order and the non-collinear G-

type antiferromagnetic order, which is induced by the DM interaction, offers a more complex way to manipulate the electric property and magnetic property by each other^{35,36}. That is, the reduction of the ferroelectric order contributed to the SHG (Fig. 2i-k) can be attributed to the orientation varying of the electric polarization³⁷. Nevertheless, we think whether and how (if yes) the magnetic field affects the strength of the polarization for BFO is still an open question and need to be further studied by all means.” We have added three new references paper as Ref. 33, Ref.34, and Ref. 36 in our revised manuscript.

References

17. Heron JT, *et al.* Deterministic switching of ferromagnetism at room temperature using an electric field. *Nature* **516**, 370-373 (2014).
28. Moriya T. Anisotropic Superexchange Interaction and Weak Ferromagnetism. *Physical Review* **120**, 91-98 (1960).
29. Cheong S-W, Mostovoy M. Multiferroics: a magnetic twist for ferroelectricity. *Nat Mater* **6**, 13-20 (2007).
30. Pan H, *et al.* Ultrahigh energy storage in superparaelectric relaxor ferroelectrics. *Science* **374**, 100-104 (2021).
31. Yao H, Guo E-J, Ge C, Wang C, Yang G, Jin K. Photon-interactions with perovskite oxides. *Chinese Physics B* **31**, 088106 (2022).
33. Lee S, *et al.* Negative magnetostrictive magnetoelectric coupling of BiFeO₃. *Physical Review B* **88**, 060103 (2013).
34. Li M-R, *et al.* Magnetostriction-polarization coupling in multiferroic Mn₂MnWO₆. *Nature Communications* **8**, 2037 (2017).
35. Tokunaga M, *et al.* Magnetic control of transverse electric polarization in BiFeO₃. *Nature Communications* **6**, 5878 (2015).
36. Bordács S, *et al.* Magnetic Field Control of Cycloidal Domains and Electric Polarization in Multiferroic BiFeO₃. *Phys Rev Lett* **120**, 147203 (2018).
37. Kimura T, Goto T, Shintani H, Ishizaka K, Arima T, Tokura Y. Magnetic control of ferroelectric polarization. *Nature* **426**, 55-58 (2003).

Again, we are very grateful for this reviewer’s insightful comments and constructive suggestions, which have helped us greatly in improving the quality of our manuscript.

Reviewer #2:

Main Comment. The authors report magnetoelectric coupling phenomena probed by wide temperature-range second harmonic generation (WT-SHG), and show the manipulation of the ferroelectric and antiferromagnetic orders in epitaxial and freestanding BiFeO₃ films. Developing RA-SHG technique is a unique way to investigate magnetoelectric coupling for other freestanding multiferroic films or 2D materials. The manuscript has the potential to be published in NC, but there are still many issues in the present form. The authors are suggested to respond following comments/questions:

Response. *We are grateful for the positive consideration from this referee. We thank the reviewer for the constructive suggestions that are important for the improvement of the manuscript. We have addressed the reviewer's suggestion in the following and in the revised manuscript.*

Comment 1. The authors need to explain why they chose BFO film with a thickness of 47nm? BFO has special phase at such thickness? BFO has a special stress state at such thickness? BFO has greater polarization or larger optical second harmonic effects at such thickness?

Response 1. *Thanks for the reviewer's valuable comments on our manuscript. The BFO films were chosen with a thickness of 47 nm because we want to prepare large-scale freestanding BFO films and well strained BFO/STO films with the same thickness. If the thickness of BFO films was too small, it will be unfeasible to obtain the large-scale freestanding BFO film. On the other hand, if the thickness was too large, BFO/STO films would relax too much on the upper surface, which is not conducive to perform the contrast experiments before and after stress releasing.*

By following the reviewer's suggestion, we have also addressed the phase of BFO films and added the following sentences on page 3 of the revised manuscript: "From the measurements of XRD (Fig. S1a), RSM (Fig. S1g-l), FFT patterns (Fig. S3), and RA-SHG, it can be concluded that: with the thickness of 47 nm, the stress on the BFO

films from the substrates was well kept, so that the epitaxial BFO films on STO and those on SAO/STO were in a tetragonal-like (T-like) phase and most likely with a P4mm space group, consistent with our previous study¹⁹, while it was also feasible to obtain the large-scale freestanding BFO film which stayed in a rhombohedral-like (R-like) phase and with an R3c space group.” We have added one new reference paper as Ref. 19 in our revised manuscript.

References

19. Wang J-s, *et al.* Evolution of structural distortion in BiFeO₃ thin films probed by second-harmonic generation. *Sci Rep* **6**, 38268 (2016).

Comment 2. The authors calculated the polarization magnitude of BFO/STO, BFO/SAO/STO, and freestanding BFO from HAADF images in Fig. 1, and they made a simple comparison among these three samples. The authors should quantitatively explain the polarization magnitude for these three samples, and compare the polarization intensity with reported literature.

Response 2. *We are very grateful to the reviewer for the constructive suggestions on our manuscript. We fully agree with the reviewer’s suggestion. In the revised manuscript, on page 3 of the main text, we have added the paragraph: “To quantitatively evaluate the polarization evolution in these three samples, analysis at unit cell scale is needed. An empirical linear relationship^{22, 23} between the polarization strength with respect to the offset between the Fe cation and the four surrounding Bi cations was adopted to obtain the polarization semi-quantitatively for BFO/STO, BFO/SAO/STO, and freestanding BFO films. The profile of polarization (Fig. 1j), also denoted by the yellow vectors in Fig. 1g-i, suggests that the maximum amplitude observed in BFO/STO is about 94.92 $\mu\text{C}/\text{cm}^2$, which is consistent with the results of 60-100 $\mu\text{C}/\text{cm}^2$ reported in the literature^{24, 25, 26}, and it slightly decreased to about 87.30 $\mu\text{C}/\text{cm}^2$ with the SAO buffer layer, while the minimum one appeared in the freestanding BFO films is about 80.03 $\mu\text{C}/\text{cm}^2$. The reduction of polarization amplitude in BFO/STO, BFO/SAO/STO, and freestanding BFO, consistent with the*

results reported in the literature²⁷, is due to the sequential relaxation of the strain. The polarization magnitude calculated from TEM method is in good agreement with that from SHG (Fig. 1k), confirming the validity of our results.”. We have added four new references paper as Ref. 24, Ref.25, Ref.26, and Ref. 27 in our revised manuscript.

References

22. Abrahams SC, Kurtz SK, Jamieson PB. Atomic Displacement Relationship to Curie Temperature and Spontaneous Polarization in Displacive Ferroelectrics. *Physical Review* **172**, 551-553 (1968).
23. Nelson CT, *et al.* Spontaneous Vortex Nanodomain Arrays at Ferroelectric Heterointerfaces. *Nano Lett* **11**, 828-834 (2011).
24. Wang J, *et al.* Epitaxial BiFeO₃ Multiferroic Thin Film Heterostructures. *Science* **299**, 1719-1722 (2003).
25. Neaton JB, Ederer C, Waghmare UV, Spaldin NA, Rabe KM. First-principles study of spontaneous polarization in multiferroic BiFeO₃. *Physical Review B* **71**, 014113 (2005).
26. Lebeugle D, Colson D, Forget A, Viret M. Very large spontaneous electric polarization in BiFeO₃ single crystals at room temperature and its evolution under cycling fields. *Appl Phys Lett* **91**, 022907 (2007).
27. Shi Q, *et al.* The role of lattice dynamics in ferroelectric switching. *Nature Communications* **13**, 1110 (2022).

Comment 3. In Fig. 2b and Fig. 2g, the authors present the SHG intensity of BFO films as a function of temperature and temperature dependent SHG signals contributed by antiferromagnetic order respectively. The temperature dependent SHG signals contributed by ferroelectric order should be provide and analysis.

Response 3. *Thanks for the reviewer’s valuable suggestions on our manuscript. We fully agree with the reviewer’s comments. We have added a new figure showing the temperature dependent SHG signal contributed by ferroelectric order for BFO/STO films, as Figure S6 and below.*

Figure S6: Temperature-dependent SHG signal contributed by ferroelectric order extracted from the RA-SHG results of BFO/STO films at different temperatures.

The following sentences have been added on page 4 of the revised manuscript: “The temperature-dependent SHG signal (extracted from Fig. 2c-f and Fig. S5) contributed by ferroelectric order gradually decreases with the increase of temperature (Fig. S6), indicating that the ferroelectric properties are weakening with the increase of temperature, which is consistent with the ferroelectric order parameter variation studied by XRD¹³.” We have added one new figure as Fig. S6 in our revised supplementary materials.

References

- Infante IC, *et al.* Bridging Multiferroic Phase Transitions by Epitaxial Strain in BiFeO₃. *Phys Rev Lett* **105**, 057601 (2010).

Comment 4. The authors claim that “the strongest (or weakest) ferroelectric order in BFO/STO (or freestanding BFO) can be obtained while the antiferromagnetic order is the strongest (or the weakest), based on the results shown in Fig. 3f-g and Fig. 2”. For comparison, the authors are suggested to provide temperature-dependent SHG signals contributed by ferroelectric order and magnetic field-dependent SHG signals contributed by antiferromagnetic order for BFO/STO, BFO/SAO/STO, and freestanding BFO films.

Response 4. We greatly appreciate the reviewer’s constructive suggestions. We have provided the temperature-dependent SHG signal contributed by the ferroelectric order of BFO/STO films in Fig. S6. Combining that with Fig. 2b-g, it can be found that: Below the Néel temperature (T_N), the strongest (or weakest) ferroelectric order in BFO/STO can be obtained when the antiferromagnetic order is the strongest (or the weakest). While above the T_N , only the SHG signal contributed by the ferroelectric order persists.

We have added a new figure showing the magnetic field-dependent SHG signal contributed by antiferromagnetic order for freestanding BFO films in Figure S11a and below.

Figure S11a: Magnetic field-dependent SHG signal contributed by antiferromagnetic order extracted from the RA-SHG results of freestanding BFO films at different magnetic fields.

The result shows that the magnetic field-dependent SHG signal contributed by antiferromagnetic order also exhibits an up-set down parabolic behavior. The following quoted sentences have been added on page 6 in the revised manuscript: “The magnetic field-dependent SHG signal (extracted from Fig. 2h-k) contributed by antiferromagnetic order and ferroelectric order also exhibits an up-set down parabolic behavior (Fig. S11).” We have added a new figure as Fig. S11a in our revised supplementary materials. In addition, we preliminarily studied the temperature-dependent SHG signal contributed by ferroelectric order for BFO/SAO/STO (and

freestanding BFO) and the magnetic field-dependent SHG signal contributed by antiferromagnetic order for BFO/STO (and BFO/SAO/STO), and found they behaved similarly with those in Fig. S6 and S11a, respectively, and further systematic study is planned. We sincerely appreciate the reviewer for the inspiration!

Comment 5. How does the ferroelectric as a function of applied magnetic field for differently strained BFO films?

Response 5. *Thanks again for the reviewer’s very constructive suggestion. We have added a new figure showing the magnetic field-dependent SHG signal contributed by ferroelectric order for freestanding BFO films in Figure S11b and below.*

Figure S11b: Magnetic field-dependent SHG signals contributed by ferroelectric order extracted from the RA-SHG results of freestanding BFO films at different magnetic fields.

The following quoted sentences have been added on page 6 in the revised manuscript: “The magnetic field-dependent SHG signal (extracted from Fig. 2h-k) contributed by antiferromagnetic order and ferroelectric order also exhibits an upset down parabolic behavior (Fig. S11).” In addition, we have preliminarily found the magnetic field-dependent SHG signal contributed by ferroelectric order for BFO/STO (and BFO/SAO/STO) seems similar with that in S11b, and further systematic study is planned. We deeply appreciate the reviewer’s inspiration.

Comment 6. As the magnetoelectric-coupling was suppressed by strain releasing, as well as the antiferromagnetic and ferroelectric orders, how does the SHG patterns change under strain manipulate for freestanding BFO films?

Response 6. *We are very grateful for the reviewer's constructive comments. To see the variation of SHG under strain manipulation for freestanding BFO films, the freestanding BFO films were transferred onto the flexible polydimethylsiloxane (PDMS) and set on our self-designed continuous-stretching optical platform. When we performed SHG measurements on the samples, we found that there was a ring of light around the reflected light spot, which leads to fairly large noise in the measured data. We suspect this noise was somehow caused by the PDMS. In addition, we found that the surface roughness of PDMS is relatively large through optical microscopy, which may also be the reason for the large noise in the measurement data. In the future, we will definitely try to find something to replace PDMS or find some other way to solve this problem in our further study. We sincerely appreciate the reviewer for the inspiration!*

Then we have tried to measure the freestanding BFO films with continuous in-plane uniaxial strain and using a far-field transmission geometry light path (in order to focus on in-plane polarization changes), the results are shown below:

Figure R1: *Variation of RA-SHG patterns of the freestanding BFO films with*

increasing in-plane uniaxial strain.

We can see that with the increasing of the in-plane uniaxial strain, the maximum value of the SHG signal of the freestanding BFO films is also increasing gradually, which indicates that the in-plane polarization is increasing. This is consistent with the results reported in the literature that the polarization of the freestanding BFO film will rotate from the out-of-plane direction to the in-plane direction gradually^{R1} under in-plane uniaxial tensile stress. Besides, with the applying of in-plane stress, the shape of the RA-SHG pattern is also continuously changing. Those results roughly reflect that the magnetoelectric coupling of the freestanding BFO films will also change under the applying of different strains. However, as these results are not comparable with those presented in our manuscript due to the difference between the different geometry of light path, we didn't add these results into the manuscript.

References

- R1. Zang Y, et al. Giant Thermal Transport Tuning at a Metal/Ferroelectric Interface. *Adv Mater* **34**, 2105778 (2022).

Again, we appreciate the inspiring comments and constructive suggestions from this reviewer very much, which helped us a lot in improving our manuscript.

Reviewer #3:

Main Comment. The authors intensively studied the magnetoelectric feature of thin-film BiFeO₃ in both strained and freestanding film forms, SHG is applied to intrinsically probe their tuned multiferroics under multiple coupling of magnetism, polarization, and optical response. These findings are expected to unveil the magnetoelectric coupling mechanism of related materials.

Response. *We greatly appreciate the reviewer for the positive recommendation and valuable comments. We thank the reviewer for the constructive suggestions that are important for the improvement of the manuscript. We have addressed the reviewer's suggestion in the following and in the revised manuscript.*

Comment 1. The symmetry of film form BiFeO₃ is not addressed. Is that the rhombohedral R3c or tetragonal P4mm? Although all can be related to the pseudo-cubic structure, it is necessary to clarify this since different polymorphs of BFO have been reported in thickness/substrate dependent cases.

Response 1. *Thanks for the reviewer's valuable comments on our manuscript. We fully agree with the reviewer's suggestion. The XRD (Fig. S1a), RSM (Fig. S1g-l), fast fourier transform (FFT) patterns (Fig. S3), and RA-SHG (Fig. 3f-g) measurements show that the epitaxial BFO films are in a tetragonal-like (T-like) phase and most likely with a P4mm space group, consistent with our previous study¹⁹, while the freestanding BFO films are in a rhombohedral-like (R-like) phase and with an R3c space group.*

In the revised manuscript, the following quoted sentences have been added on page 3: "From the measurements of XRD (Fig. S1a), RSM (Fig. S1g-l), FFT patterns (Fig. S3), and RA-SHG, it can be concluded that: with the thickness of 47 nm, the stress on the BFO films from the substrates was well kept, so that the epitaxial BFO films on STO and those on SAO/STO were in a tetragonal-like (T-like) phase and most likely with a P4mm space group, consistent with our previous study¹⁹, while it was also feasible to obtain the large-scale freestanding BFO film which stayed in a

rhombohedral-like (R-like) phase and with an R3c space group.” We have added one new reference paper as Ref. 19 in our revised manuscript.

References

19. Wang J-s, *et al.* Evolution of structural distortion in BiFeO₃ thin films probed by second-harmonic generation. *Sci Rep* **6**, 38268 (2016).

Comment 2. The cell evolution indicates that the lattice mismatch between the substrate and BFO introduce tensile stress to BFO since the dimension along the c-axis become smaller in freestanding state, says from 4.061 to 3.979 Å? Is this for the unit cell parameter c of the P4mm symmetry? Or the d value for certain (hkl)?

Response 2. *Thanks for the reviewer’s valuable comments on our manuscript. The c-axis lattice constants indeed refer to the unit cell parameter c, namely $d_{(001)}$. As mentioned in response 1, these values (4.061 and 3.979 Å) of $d_{(001)}$ are of the P4mm symmetry for the BFO epitaxially grown on SAO/STO and of the R3c symmetry for the free standing one, respectively.*

Comment 3. As far as I understood, structurally, the freestanding and strained BFO should be considered as different phases due to their large dimensional difference, since the magnetostriction-polarization coupling is commonly observed in related system, such as in Mn₂MnWO₆ (Nat. Commun., 2017, 8, 2037). So the multiple coupling effect cannot be simply counted when spin-lattice-phonon-dipole are got involved. So it could be a plus to measure any samples before and after removal of the intermediate Sr-Al-O layer for comparison.

Response 3. *Thanks for the reviewer’s valuable suggestions on our manuscript. By following the reviewer’s inspiring suggestion, we have revised the relevant discussion in the revised manuscript on page 5 as: “The magnetoelectric coupling in BFO was induced by its intrinsic and significant spin-orbital coupling. The antiferromagnetic order would introduce additional electric polarization via spin-orbital coupling directly, which is well-known as magneto-striction phenomenon^{33,34}. Furthermore, the*

coupling between the ferroelectric order and the non-collinear G-type antiferromagnetic order, which is induced by the DM interaction, offers a more complex way to manipulate the electric property and magnetic property by each other^{35,36}.” We have added three new references paper as Ref. 33, Ref.34 (*Nat. Commun.*, 2017, **8**, 2037), and Ref. 36 in our revised manuscript.

We performed the XRD, RSM (Fig. S1), and TEM characterization of the BFO samples before and after removing the intermediate SAO layer for comparison (Fig. 1e-f on page 14). The results show that the lattice and structure of the samples changed after removing the intermediate SAO layer. The effects of these changes on the ferroelectricity, weak ferromagnetism, and magnetoelectric coupling of BFO films are illustrated by SHG measurements with varying magnetic fields (Fig. 3i-k on page 16) before and after removing the intermediate SAO layer. By following the reviewer’s suggestion we have also clarified the variation of BFO phase before and after removing the intermediate SAO layer in our revised manuscript by adding the following sentences on page 3: “From the measurements of XRD (Fig. S1a), RSM (Fig. S1g-l), FFT patterns (Fig. S3), and RA-SHG, it can be concluded that: with the thickness of 47 nm, the stress on the BFO films from the substrates was well kept, so that the epitaxial BFO films on STO and those on SAO/STO were in a tetragonal-like (T-like) phase and most likely with a P4mm space group, consistent with our previous study¹⁹, while it was also feasible to obtain the large-scale freestanding BFO film which stayed in a rhombohedral-like (R-like) phase and with an R3c space group.” We have added one new reference paper as Ref. 19 in our revised manuscript.

References

33. Lee S, *et al.* Negative magnetostrictive magnetoelectric coupling of BiFeO₃. *Physical Review B* **88**, 060103 (2013).
34. Li M-R, *et al.* Magnetostriction-polarization coupling in multiferroic Mn₂MnWO₆. *Nature Communications* **8**, 2037 (2017).
35. Tokunaga M, *et al.* Magnetic control of transverse electric polarization in BiFeO₃. *Nature Communications* **6**, 5878 (2015).
36. Bordács S, *et al.* Magnetic Field Control of Cycloidal Domains and Electric Polarization in Multiferroic BiFeO₃. *Phys Rev Lett* **120**, 147203 (2018).

19. Wang J-s, *et al.* Evolution of structural distortion in BiFeO₃ thin films probed by second-harmonic generation. *Sci Rep* **6**, 38268 (2016).

Comment 4. The SHG was detected by reflection mode. In the strained BFO film, the out-plane stress reaches maximum in the BFO and substrate interface, being significantly faded with growing film thickness. My question is, for the ~50 nm thickness film, what is the stress effect on the upper surface compared with the freestanding one? So it is better to evaluate strain effect on surfaces of both strained and freestanding (top and bottom) ones by the lattice dimension. Presumably, the thicker the film, the smaller the lattice dimension of the outer-plane, the limit should be equal to that of the freestanding case.

Response 4. *Thanks for the reviewer's valuable comments on our manuscript. We fully agree with the reviewer's comment and presumption. Given that the lattice constant of the BFO bulk and STO substrate are 3.965 Å and 3.905 Å respectively, the STO substrate exerts an in-plane compressive stress on the BFO. By following the reviewer's suggestion, we have added a new figure as Fig. S3 in the revised manuscript, and the following quoted sentences have been added on page 2 of the main text: "From the HAADF-STEM images and fast fourier transform (FFT) patterns on the upper surface and at the interface of BFO grown on STO (Fig. S3) respectively, we find that the in-plane lattice constant is basically the same (3.89 Å) on the upper surface with that near the interface, while the out-of-plane lattice constant is a little smaller (4.02 Å) on the upper surface than that (4.09 Å) close to the interface between BFO and the substrate of STO. Although there is some relaxation of the compressive stress from the substrates for the lattices on the upper surface of BFO films with the thickness of 47 nm epitaxially grown on the STO, the stress was well kept within the films concluded from the larger out of plane lattice constant than that (3.979 Å) of freestanding BFO (Fig. S1a)." We have added one new figure as Fig. S3 in our revised supplementary materials.*

Figure S3: a Atomic-scale HAADF-STEM images of the BFO/STO films. b and c Fast fourier transform (FFT) patterns of the bottom BFO and the top BFO, respectively.

Comment 5. Is the freestanding film totally relaxed? Any SHG difference measured on the top and bottom (the side epitaxially grown on substrate) surfaces?

Response 5. *Thanks for the reviewer's valuable comments on our manuscript. We think the freestanding BFO films are totally relaxed, as we can confirm that the sacrificial layer SAO is completely ablated in deionized water, that is, the BFO films and the STO substrate can be completely separated. From Figure R2a below, we can see that the freestanding BFO films were completely separated from the substrate and floated on the surface of deionized water.*

*We have added a detailed description of the release and transfer process of the freestanding BFO films in the **Methods** section of the revised manuscript on page 8: "A 10 mm × 20 mm × 0.5 mm transparent and flexible PDMS was tightly covered on the surface of the BFO/SAO/STO epitaxial films. They were then immersed in*

deionized water at room temperature for about 30 minutes until the SAO sacrificial layer was completely dissolved and the BFO films were separated from the STO (001) substrate. After that, the PDMS together with the freestanding BFO films we dried with N₂ gas for several minutes. Then, it was transferred onto any desired substrate (such as a silicon wafer or a TEM grid), with the entire stacking annealed at 90 °C for 30 minutes to promote adhesion. After cooling to 70°C and slowly peeling off the PDMS with tweezers, the transferred films on an arbitrary substrate were obtained.”

Following the reviewer’s suggestion, we transferred the freestanding BFO films onto STO substrates with the bottom side and the top side on the surface, respectively. Then we performed RA-SHG measurements on the samples and the results as shown below:

Figure R2: *a* Dissolving the sacrificial layer with deionized water to obtain freestanding BFO films. *b* RA-SHG measurements on the top side, the top side, and the bottom side of the freestanding BFO films.

From the results above, we can see that the shape of the SHG patterns on the top side and the bottom side are basically the same.

Comment 6. The SHG-magnetism-polarization-lattice coupling is rather complicated. The authors are brave enough to touch this. It will be better to calculate the spontaneous polarization according the cell-parameter-based cif files by either first-principles calculations or point-charge model, so that one can tell the structurally

dipole contribution.

Response 6. *Thank you for the inspiring comments. We fully agree that SHG-magnetism-polarization-lattice coupling is challenging for us, which, on the other hand, attracts and excited us while this work was proceeding. On page 10 in the revised supplementary materials, we have added one new table and replaced the related paragraphs and figure by the following new ones:*

“As shown in Fig. S8, we have calculated the spontaneous polarization of BFO with intrinsic FM order and G-type AFM order, respectively. The lattice constants were fixed as our XRD and TEM experimental results of the STO-BFO during our first-principles calculations ($a=b=3.89 \text{ \AA}$, $c=4.06 \text{ \AA}$). From Table S1 and Fig. S8, we can see that the electronic dipole and ionic dipole are both larger in BFO with G-type antiferromagnetic order than those with the ferromagnetic order in all three directions, namely a ([100]), b ([010]), and c ([001]). The enlargement of polarization demonstrates theoretically that the antiferromagnetic order would enhance the polarization.

Table S1: Ionic and electronic polarization of BFO ($a=b=3.89 \text{ \AA}$, $c=4.06 \text{ \AA}$) in the directions a, b, and c, respectively.

	$P_{ion.,a}$	$P_{ion.,b}$	$P_{ion.,c}$	$P_{ele.,a}$	$P_{ele.,b}$	$P_{ele.,c}$
	$ e \cdot \text{\AA}$	$ e \cdot \text{\AA}$	$ e \cdot \text{\AA}$	$ e \cdot \text{\AA}$	$ e \cdot \text{\AA}$	$ e \cdot \text{\AA}$
G-AFM	-149.3	-149.3	-155.7	0.547	0.547	-2.681
FM	-149.2	-149.2	-155.4	0.443	0.443	-2.329

Fig. S8: First-principles calculation results of $2 \times 2 \times 2$ BFO supercell under different magnetic structures. The absolute value of the electronic dipole moment (in

blue) and the ionic dipole moment (in red) of BFO with ferromagnetic (FM) order (hollow) and G-type antiferromagnetic (G-AFM) order (filled) in directions a, b, and c.”

We have added one new table as Table S1 on page 10 in our revised supplementary materials.

Comment 7. The properties of these kind of films are very sensitive to defect, such as oxygen vacancy. Any characteristics on this – says how reproducible of the samples from different batches?

Response 7. *Thanks for the reviewer’s valuable comments on our manuscript. It is true that the performance of this type of oxide films is very sensitive to their oxygen vacancy, which is most easily affected by the growth oxygen pressure, growth temperature, and laser power. In order to illustrate the influence of oxygen vacancy on the properties of samples, we supplemented the SHG results of BFO films grown under different oxygen pressures (2.5 Pa, 5 Pa, 10 Pa, and 20 Pa) as shown in Figure R3 and below:*

Figure R3: *The SHG measurements of BFO films grown under different oxygen pressures (2.5 Pa, 5 Pa, 10 Pa, and 20 Pa).*

We can find out that the effect of oxygen vacancy on oxide films is very obvious. Under low growth oxygen pressure, the BFO oxide films are likely to generate more oxygen vacancies, which reduces the polarity, and the corresponding SHG intensity is

lower.

In order to suppress the formation of oxygen vacancy, in this work, we generally grow samples under high oxygen pressure (20 Pa) and annealed them in situ for 10 minutes. So that the concentration of oxygen vacancy for the samples will be minimized and the distribution will be more uniform. As each batch of samples was grown in the same growth conditions, the repeatability is quite good. The SHG measurements of different batches of BFO films are shown in Figure R4 and below:

Figure R4: *The SHG measurements of different batches of BFO films (both under the oxygen pressure of 20 Pa).*

Again, we appreciate the inspiring comments and constructive suggestions from this reviewer very much, which helped us a lot to improve our manuscript.

List of changes to the manuscript

1. *In the revised manuscript on page 3-4, we have added the relevant description as: “It has been found that the ferroelectric polarization in the BFO films directly coupled with the non-collinear G-type antiferromagnetic as well as the weak ferromagnetic moment driven by the Dzyaloshinskii-Moriya (DM) interaction, which arises from spin-orbital coupling in antisymmetric systems^{17, 28, 29, 30, 31}”.*

2. *On page 5, we have rewritten the relevant description as: “The magnetoelectric coupling in BFO was induced by its intrinsic and significant spin-orbital coupling. The antiferromagnetic order would introduce additional electric polarization via spin-orbital coupling directly, which is well-known as magneto-striction phenomenon^{33,34}. Furthermore, the coupling between the ferroelectric order and the non-collinear G-type antiferromagnetic order, which is induced by the DM interaction, offers a more complex way to manipulate the electric property and magnetic property by each other^{35,36}. That is, the reduction of the ferroelectric order contributed to the SHG (Fig. 2i-k) can be attributed to the orientation varying of the electric polarization³⁷. Nevertheless, we think whether and how (if yes) the magnetic field affects the strength of the polarization for BFO is still an open question and need to be further studied by all means.” We have added three new references paper as Ref. 33, Ref.34 (Nat. Commun., 2017, 8, 2037), and Ref. 36 in our revised manuscript.*

3. *we have also addressed the phase of BFO films and added the following sentences on page 3 of the revised manuscript: “From the measurements of XRD (Fig. S1a), RSM (Fig. S1g-l), FFT patterns (Fig. S3), and RA-SHG, it can be concluded that: with the thickness of 47 nm, the stress on the BFO films from the substrates was well kept, so that the epitaxial BFO films on STO and those on SAO/STO were in a tetragonal-like (T-like) phase and most likely with a P4mm space group, consistent with our previous study¹⁹, while it was also feasible to obtain the large-scale freestanding BFO film which stayed in a rhombohedral-like (R-like) phase and with an R3c space group.” We have added one new reference paper as Ref. 19 in our*

revised manuscript.

4. *In the revised manuscript, on page 3 of the main text, we have added the paragraph: “To quantitatively evaluate the polarization evolution in these three samples, analysis at unit cell scale is needed. An empirical linear relationship^{22, 23} between the polarization strength with respect to the offset between the Fe cation and the four surrounding Bi cations was adopted to obtain the polarization semi-quantitatively for BFO/STO, BFO/SAO/STO, and freestanding BFO films. The profile of polarization (Fig. 1j), also denoted by the yellow vectors in Fig. 1g-i, suggests that the maximum amplitude observed in BFO/STO is about $94.92 \mu\text{C}/\text{cm}^2$, which is consistent with the results of $60\text{-}100 \mu\text{C}/\text{cm}^2$ reported in the literature^{24, 25, 26}, and it slightly decreased to about $87.30 \mu\text{C}/\text{cm}^2$ with the SAO buffer layer, while the minimum one appeared in the freestanding BFO films is about $80.03 \mu\text{C}/\text{cm}^2$. The reduction of polarization amplitude in BFO/STO, BFO/SAO/STO, and freestanding BFO, consistent with the results reported in the literature²⁷, is due to the sequential relaxation of the strain. The polarization magnitude calculated from TEM method is in good agreement with that from SHG (Fig. 1k), confirming the validity of our results.”. We have added four new references paper as Ref. 24, Ref.25, Ref.26, and Ref. 27 in our revised manuscript.*

5. *The following sentences have been added on page 4 of the revised manuscript: “The temperature-dependent SHG signal (extracted from Fig. 2c-f and Fig. S5) contributed by ferroelectric order gradually decreases with the increase of temperature (Fig. S6), indicating that the ferroelectric properties are weakening with the increase of temperature, which is consistent with the ferroelectric order parameter variation studied by XRD¹³.” We have added one new figure as Fig. S6 in our revised supplementary materials.*

6. *The following quoted sentences have been added on page 6 in the revised manuscript: “The magnetic field-dependent SHG signal (extracted from Fig. 2h-k) contributed by antiferromagnetic order and ferroelectric order also exhibits an upset*

down parabolic behavior (Fig. S11).” We have added a new figure as Fig. S11 in our revised supplementary materials.

7. We have added a new figure as Fig. S3 in the revised manuscript, and the following quoted sentences have been added on page 2 of the main text: “From the HAADF-STEM images and fast fourier transform (FFT) patterns on the upper surface and at the interface of BFO grown on STO (Fig. S3) respectively, we find that the in-plane lattice constant is basically the same (3.89 Å) on the upper surface with that near the interface, while the out-of-plane lattice constant is a little smaller (4.02 Å) on the upper surface than that (4.09 Å) close to the interface between BFO and the substrate of STO. Although there is some relaxation of the compressive stress from the substrates for the lattices on the upper surface of BFO films with the thickness of 47 nm epitaxially grown on the STO, the stress was well kept within the films concluded from the larger out of plane lattice constant than that (3.979 Å) of freestanding BFO (Fig. S1a).” We have added one new figure as Fig. S3 in our revised supplementary materials.

8. We have added a detailed description of the release and transfer process of the freestanding BFO films in the **Methods** section of the revised manuscript on page 8: “A 10 mm × 20 mm × 0.5 mm transparent and flexible PDMS was tightly covered on the surface of the BFO/SAO/STO epitaxial films. They were then immersed in deionized water at room temperature for about 30 minutes until the SAO sacrificial layer was completely dissolved and the BFO films were separated from the STO (001) substrate. After that, the PDMS together with the freestanding BFO films we dried with N₂ gas for several minutes. Then, it was transferred onto any desired substrate (such as a silicon wafer or a TEM grid), with the entire stacking annealed at 90 °C for 30 minutes to promote adhesion. After cooling to 70°C and slowly peeling off the PDMS with tweezers, the transferred films on an arbitrary substrate were obtained.”

9. On page 10 in the revised supplementary materials, we have added one new table and replaced the related paragraphs and figure.

References

13. Infante IC, *et al.* Bridging Multiferroic Phase Transitions by Epitaxial Strain in BiFeO₃. *Phys Rev Lett* **105**, 057601 (2010).
17. Heron JT, *et al.* Deterministic switching of ferromagnetism at room temperature using an electric field. *Nature* **516**, 370-373 (2014).
19. Wang J-s, *et al.* Evolution of structural distortion in BiFeO₃ thin films probed by second-harmonic generation. *Sci Rep* **6**, 38268 (2016).
22. Abrahams SC, Kurtz SK, Jamieson PB. Atomic Displacement Relationship to Curie Temperature and Spontaneous Polarization in Displacive Ferroelectrics. *Physical Review* **172**, 551-553 (1968).
23. Nelson CT, *et al.* Spontaneous Vortex Nanodomain Arrays at Ferroelectric Heterointerfaces. *Nano Lett* **11**, 828-834 (2011).
24. Wang J, *et al.* Epitaxial BiFeO₃ Multiferroic Thin Film Heterostructures. *Science* **299**, 1719-1722 (2003).
25. Neaton JB, Ederer C, Waghmare UV, Spaldin NA, Rabe KM. First-principles study of spontaneous polarization in multiferroic BiFeO₃. *Physical Review B* **71**, 014113 (2005).
26. Lebeugle D, Colson D, Forget A, Viret M. Very large spontaneous electric polarization in BiFeO₃ single crystals at room temperature and its evolution under cycling fields. *Appl Phys Lett* **91**, 022907 (2007).
27. Shi Q, *et al.* The role of lattice dynamics in ferroelectric switching. *Nature Communications* **13**, 1110 (2022).
28. Moriya T. Anisotropic Superexchange Interaction and Weak Ferromagnetism. *Physical Review* **120**, 91-98 (1960).
29. Cheong S-W, Mostovoy M. Multiferroics: a magnetic twist for ferroelectricity. *Nat Mater* **6**, 13-20 (2007).
30. Pan H, *et al.* Ultrahigh energy storage in superparaelectric relaxor ferroelectrics. *Science* **374**, 100-104 (2021).
31. Yao H, Guo E-J, Ge C, Wang C, Yang G, Jin K. Photon-interactions with perovskite oxides. *Chinese Physics B* **31**, 088106 (2022).
33. Lee S, *et al.* Negative magnetostrictive magnetoelectric coupling of BiFeO₃. *Physical Review B* **88**, 060103 (2013).
34. Li M-R, *et al.* Magnetostriction-polarization coupling in multiferroic Mn₂MnWO₆. *Nature Communications* **8**, 2037 (2017).
35. Tokunaga M, *et al.* Magnetic control of transverse electric polarization in BiFeO₃. *Nature Communications* **6**, 5878 (2015).
36. Bordács S, *et al.* Magnetic Field Control of Cycloidal Domains and Electric Polarization in Multiferroic BiFeO₃. *Phys Rev Lett* **120**, 147203 (2018).
37. Kimura T, Goto T, Shintani H, Ishizaka K, Arima T, Tokura Y. Magnetic control of ferroelectric polarization. *Nature* **426**, 55-58 (2003).

REVIEWERS' COMMENTS

Reviewer #1 (Remarks to the Author):

The authors provided satisfactory responses to all of the comments presented by the reviewers. They also modified the paper accordingly. I recommend the publication of the paper and I trust the authors will carefully proofread and cross-check the information in the text.

Reviewer #2 (Remarks to the Author):

The authors have addressed my comments properly. I would recommend to publish the manuscript as it is.

Reviewer #3 (Remarks to the Author):

The revised manuscript is significantly improved, and can be published after minor correction on some typos, the variables should be italic, the space group symbol should be addressed in standard way, there should be a space between a number and its unit, and so on.

RESPONSE TO REVIEWER COMMENTS

Dear Referees,

Thank you very much for your recent comments concerning our manuscript (NCOMMS-23-01445A) entitled “Magnetolectric Coupling in Multiferroics Probed by Optical Second Harmonic Generation”. We thank all the reviewers for positive recommendation and valuable comments regarding our research paper. Each of your insights have served to strengthen our manuscript. We have made improved revisions to the manuscript format and details.

Reviewer #1:

Comment. The authors provided satisfactory responses to all of the comments presented by the reviewers. They also modified the paper accordingly. I recommend the publication of the paper and I trust the authors will carefully proofread and cross-check the information in the text.

Response. *We greatly appreciate the positive comments from this referee. We have revised the corresponding content and format of the manuscript according to the Author Checklist.*

Reviewer #2:

Main Comment. The authors have addressed my comments properly. I would recommend to publish the manuscript as it is.

Response. *We are grateful for the positive consideration from this referee. We have revised the corresponding content and format of the manuscript according to the Author Checklist.*

Reviewer #3:

Main Comment. The revised manuscript is significantly improved, and can be published after minor correction on some typos, the variables should be italic, the space group symbol should be addressed in standard way, there should be a space between a number and its unit, and so on.

Response. *We greatly appreciate the reviewer for the positive recommendation and valuable comments. We have corrected typos and formatting in the manuscript based on the reviewer's suggestions.*

Thanks again for your comments and suggestions.

Best regards,

Kuijuan Jin, on behalf of all co-authors